# Learning Active Camera for Multi-Object Navigation

**Peihao Chen**[1,6]   **Dongyu Ji**[1]   **Kunyang Lin**[1,2]   **Weiwen Hu**[1]   **Wenbing Huang**[5]
**Thomas H. Li**[6]   **Mingkui Tan**[1,7]*   **Chuang Gan**[3,4]
[1]South China University of Technology,
[2]Pazhou Laboratory, [3]MIT-IBM Watson AI Lab, [4]UMass Amherst,
[5]Gaoling School of Artificial Intelligence, Renmin University of China,
[6]Information Technology  R&D  Innovation Center of Peking University,
[7]Key Laboratory of Big Data and Intelligent Robot, Ministry of Education,
phchencs@gmail.com, mingkuitan@scut.edu.cn

## Abstract

Getting robots to navigate to multiple objects autonomously is essential yet difficult in robot applications. One of the key challenges is how to explore environments efficiently with camera sensors only. Existing navigation methods mainly focus on fixed cameras and few attempts have been made to navigate with active cameras. As a result, the agent may take a very long time to perceive the environment due to limited camera scope. In contrast, humans typically gain a larger field of view by looking around for a better perception of the environment. How to make robots perceive the environment as efficiently as humans is a fundamental problem in robotics. In this paper, we consider navigating to multiple objects more efficiently with active cameras. Specifically, we cast moving camera to a Markov Decision Process and reformulate the active camera problem as a reinforcement learning problem. However, we have to address two new challenges: 1) how to learn a good camera policy in complex environments and 2) how to coordinate it with the navigation policy. To address these, we carefully design a reward function to encourage the agent to explore more areas by moving camera actively. Moreover, we exploit human experience to infer a rule-based camera action to guide the learning process. Last, to better coordinate two kinds of policies, the camera policy takes navigation actions into account when making camera moving decisions. Experimental results show our camera policy consistently improves the performance of multi-object navigation over four baselines on two datasets.

## 1  Introduction

In the multi-object navigation task, an intelligent embodied agent needs to navigate to multiple goal objects in a 3D environment. Typically, no pre-computed map is available and the agent needs to use a stream of egocentric observations to perceive the environment. This navigation ability is the basis for indoor robots and embodied AI. Significant recent progress on this problem can be attributed to the availability of large-scale visually rich 3D datasets [71, 7, 66, 69, 32, 29], developments in high-quality 3D simulators [71, 61, 46, 2, 30], and research on deep memory-based architectures that combine geometry and semantics for learning representations of the 3D environment [8, 68, 58, 19, 31].

Despite these advances, how to efficiently perceive the environment and locate goal objects is still an unsolved problem. Agents in current research [9, 12, 49] perceive the environment via an RGB-D camera. However, the camera is set to look forward and the range of its view is often limited. As a

---

*Corresponding author.

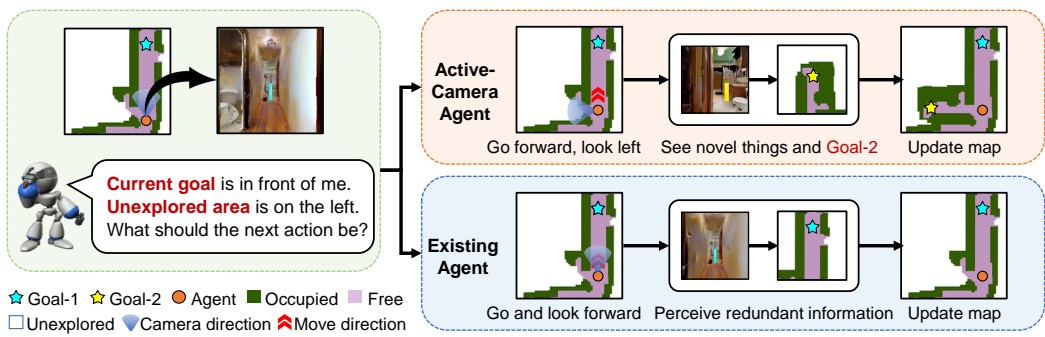

Figure 1: Illustration of an active-camera agent. The active-camera agent turns its camera to the left actively to look for novel things, *i.e., Goal-2*, when walking towards *Goal-1*.

result, these agents can only perceive the environment in front of themselves within or slightly beyond the view range [58]. As shown at the bottom-right in Figure 1, the agent keeps looking forward and perceives redundant information which is nearly identical to the previous observations. In contrast, we human beings may be attracted by surrounding things so that we turn our heads actively to receive more information when keeping walking straight. In this way, we can not only reach the position in front of ourselves efficiently but also get familiar with room arrangement and object location.

Motivated by the above observations, we design an agent which seeks to coordinate both camera and navigation actions. The agent, called **active-camera agent**, employs two kinds of policies, namely the navigation policy which determines where to go and a new camera policy which determines where to look. As shown at the top-right in Figure 1, the agent has found an object, *i.e.,* blue *Goal-1*, located at the end of a corridor. Two kinds of policies cooperate and actuate the agent to move forward and turn its camera to the left. As a result, the agent walks closer to the *Goal-1* and locates another goal, *i.e.,* yellow *Goal-2*, for the following navigation process. However, it is nontrivial to learn a good camera policy because of sophisticated RGB-D observations and complex indoor layouts. How to extract useful information from observations to judge which direction is worth being explored is difficult. Besides, the agent and the camera are moving simultaneously. How to coordinate these two actions for both the navigation and exploration processes is still unknown.

In this paper, we propose an **EXPloration-Oriented (EXPO) camera policy** to determine camera actions. Specifically, to better understand the sophisticated RGB-D observations, we transform them to a top-down occupancy map. Each pixel indicates whether it is an unexplored, free or occupied area. Such a map helps to simplify unstructured RGB-D observations and provides the necessary information, *e.g.,* explored areas and room layouts, for determining camera actions. Besides, to reduce the learning difficulty, we propose a heuristic module to infer an expected camera direction according to heuristic rules. We consider the heuristic module an expert and exploit it to guide the learning process. We then feed three types of information, *i.e.,* the progressively built map, the expected camera direction, and the upcoming navigation action, into a neural network to predict a camera action. The neural network considers both the heuristic rules and the navigation intention. We use a reinforcement learning algorithm to train the neural network by awarding camera actions that maximize the exploration area. We incorporate the EXPO camera policy with existing navigation policies [58, 68] and train them in an end-to-end manner for the multi-object navigation task. Extensive experiments on two benchmarks demonstrate the effectiveness of the proposed methods.

Our main contributions are threefold: 1) Unlike existing agents that are set to look forward, we propose a navigation paradigm that an agent coordinates camera and navigation actions for efficiently perceiving environments to solve the multi-object navigation task. 2) We propose to learn an EXPO camera policy to determine camera actions. The camera policy leverages heuristic rules to reduce the learning difficulty and takes into account the navigation intention to coordinate with the navigation policy. Such a camera policy can be incorporated with most existing navigation methods. 3) Extensive experiments demonstrate consistent improvement over four navigation methods on MatterPort3D [7] and Gibson [71] datasets. More critically, the camera policy also exhibits promising transferability to unseen scenes.

## 2  Related Work

**Visual indoor navigation.** According to the types of goal, visual navigation tasks can be categorized into different classes such as PointGoal [33, 9] task where the goal is a given coordinate, ObjectGoal [33, 68] task where the agent needs to navigate to an object given by language and ImageGoal [12, 74] task where the object is specified by an image. The location of the goal is not explicitly known in the last two categories of the above tasks and the agent is expected to explore the environment to find the goals. Classical approaches [28, 67, 34, 63, 45] solve the navigation problem via path planning [11, 34] on a constructed map, which usually requires handcraft design. Recent works [74, 53, 73, 65, 68, 15, 64, 50, 56, 38] aim to use learning-based methods in an end-to-end manner to learn a policy for navigation. Other works [9, 12, 58, 43, 47, 5, 48, 18, 52, 17, 27, 60, 23] combine the classic and learning-based methods, which use a learned SLAM module with a spatial map or topological map. We propose a camera policy, which seeks to move the camera actively and can be incorporated with existing navigation methods, to improve the navigation performance.

**Exploration for navigation.** Common methods explore the environment based on heuristics like the frontier-based exploration algorithm [72, 35, 24], which chooses a frontier point between explored and unexplored areas as the exploration goal. Recent works tackle the exploration problem via learning [9, 21, 6, 25, 22, 57, 36, 44, 54], which allows the agent to incorporate knowledge from previously seen environments and generalize to novel environments. Specially, Chen *et al.* [21] and Jayaraman *et al.* [40] use end-to-end reinforcement learning policy to maximize an external reward (*i.e.,* exploration area). Burda *et al.* [6] and Dean *et al.* [22] consider intrinsic rewards such as curiosity for efficient exploration, which performs better when external rewards are sparse. Chaplot *et al.* [9], Ramakrishnan [58] and Chen *et al.* [16] infer an exploration goal by a learned policy and navigate to it using path planner, which avoids sample complexity problem in end-to-end training. Elhafsi *et al.* [26] predict the map out of view range for better path planning. Unlike existing methods, we try to actively control the camera direction for efficient exploration.

**Active perception.** Active perception [3] aims to change the state parameters of sensors according to intelligent control strategies and gain more information about the environment. Common methods guide the view selection using information-theoretic approaches, such as mutual information [42, 13, 14, 37]. Recent work applies this strategy on different tasks such as object recognition [41, 1, 39], object localization [10, 74, 51, 55] and scene completion [40, 59]. We refer readers to [20] and [4] for a detailed review. In this paper, we propose a camera control strategy for active camera moving to help multi-object navigation.

## 3  Multi-object Navigation with Active Camera

### 3.1  Problem Formulation

Considering an agent equipped with one RGB-D camera in a novel environment, the multi-object navigation task asks the agent to explore the environment to find multiple goal objects and then navigate to them. During these processes, existing methods [68] design a navigation policy to process the observations $o$ from the environment and infer a navigation action $a^n$ (*i.e.,* FORWARD, TURN-LEFT, TURN-RIGHT, and FOUND). The navigation action is responsible for moving toward goal objects and indicates whether agents have found goals. However, these methods do not consider moving the camera direction actively during the navigation process. Thus, agents can only perceive the field along their navigation direction, which causes low efficiency in finding goal objects [58].

To resolve the above problem, we propose a new navigation paradigm that consists of a navigation policy $\pi^n(\cdot)$ and a **camera policy** $\pi^c(\cdot)$. We reformulate both the navigation and camera moving process to a Partially Observable Markov Decision Process, where the observation $o$ received by the agent does not fully specify the state of the environment. At each time step, for a given state estimated from $o$, these two policies predict a navigation action $a^n \sim \pi^n(\cdot)$ together with a camera action $a^c \sim \pi^c(\cdot)$ for active camera moving. The action space $\mathcal{A}$ for $a^n$ is similar to the navigation action space. The possible camera action includes {TURN-CAMERA-LEFT, TURN-CAMERA-RIGHT, and KEEP}, where KEEP indicates the camera direction remains unchanged. After performing the action, the policies will receive a reward $r$ whose details can be found in Section 3.3. We call the agent using this paradigm **active-camera agent** and the general scheme is shown in Figure 2.

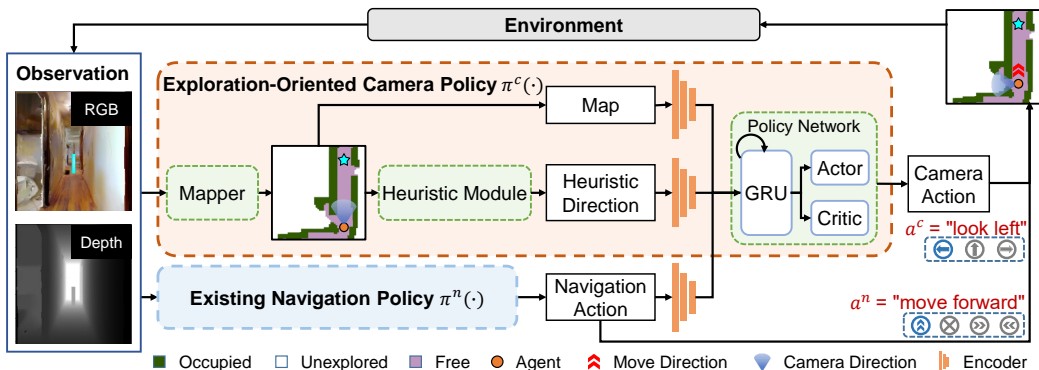

Figure 2: General scheme of an active-camera agent. The agent consists of two policies, namely the exploration-oriented camera policy determining where to look and existing navigation policy [58, 68] determining where to go.

## 3.2 Learning to Determine Camera Action

Learning a camera policy for determining camera actions is challenging because 1) it is difficult to understand complex room structure from RGB-D observations; 2) the position of agents is changing all the time so camera policies must coordinate with navigation policies. In this paper, we design an EXPO camera policy that consists of three components, *i.e.,* a mapper, a heuristic module, and a policy network. The mapper transforms the RGB-D image to an occupancy top-down map, which reveals the room layout and location of unexplored areas straightforwardly. The heuristic module infers a heuristic direction that is worth exploring. The inferred direction can be served as a reference for camera policy and reduce the learning difficulty. Then, a policy network predicts camera actions considering the encoded features of the map, heuristic direction, and upcoming navigation action. The upcoming navigation action informs the policy network of the next location of agents. The overview of the camera policy is shown in Figure 2. Next, we introduce each component in detail.

**Mapper.** We follow existing work [58] to build an occupancy map by a mapper. Each map pixel has three possible values, *i.e.,* $0, 1, 2$, indicating an unexplored, free or occupied area, respectively. Specifically, the mapper takes RGB-D as input and predicts a local map $\mathbf{M}^l$. This local map represents a small area in front of the camera. Then we integrate the local map into a global map $\mathbf{M}^g$ via a registration function $R(\cdot)$, *i.e.,* $\mathbf{M}^g = R(\mathbf{M}^l, \mathbf{M}^g, q)$, where $q$ is the camera's pose representing its position and orientation. The global map covers the whole explored environment.

**Heuristic module.** We seek to find a heuristic direction indicating which area is worth being explored by using heuristic rules. The principle is to find a direction where the camera sight is not blocked by occupancy (*e.g.,* wall). Following this principle, we draw $K$ points $\mathcal{P} = \{p_i\}_{i=1}^K$ uniformly around the agent. Each point is located at angle $\theta_i$ from the current navigation direction and $r$ meters away from the agent. We calculate geodesic distance $\mathcal{D} = \{D(p_i)\}_{i=1}^K$ from agent to these points on the progressively built global map $\mathbf{M}^g$ via A* path planning algorithm $D(\cdot)$ [34]. During path planning, we consider both the free and unexplored areas navigable. If the geodesic distance $d_i$ is smaller than a threshold $\gamma$ $(\gamma > r)$ and the point $p_i$ locates at unexplored areas in the global map, we denote these points explorable because there is no obstacle locating between the agent and point $p_i$. Moving the camera to these explorable directions allows the agent to see unexplored parts of the environment. If there exist such explorable points, we select one point closest to the camera direction and take its angle $\theta_i$ as the heuristic direction $\theta^*$. Otherwise, we set $\theta^* = 0$, *i.e.,* facing the navigation direction.

**Policy network.** With the progressively built map and inferred heuristic direction, we use a recurrent network to aggregate this information at each time step. The network takes as input map features, heuristic direction features, and navigation action features. Specifically, a convolutional neural network is exploited to extract map features from an egocentric map which is cropped from the global map. The other two types of features are encoded from the heuristic direction $\theta^*$ and navigation action $a^n$, respectively by using a learned embedding layer. The output state features from the recurrent network are fed into an actor-critic network. The actor predicts a camera action and the critic predicts the value of the agent's current state.

**Algorithm 1** Training method for active-camera agent.

**Require:** The parameters of the navigation policy $\pi^n(\cdot)$ and camera policy $\pi^c(\cdot)$, a mapper $m(\cdot)$, the number of points $K$, the radius $r$, the distance threshold $\gamma$, map registration function $R(\cdot)$, A* algorithm $D(\cdot)$, angle selection function $S(\cdot)$.
1: Initialize the parameters of $\pi^n(\cdot)$ and $\pi^c(\cdot)$ randomly.
2: Initialize an occupancy global map $\mathbf{M}^g = \mathbf{0}$.
3: **while** not convergent **do**
4:     Collect observation $o$ from environment.
5:     *// Update the global map*
6:     Let local map $\mathbf{M}^l = m(o)$ and $\mathbf{M}^g = R(\mathbf{M}^l, \mathbf{M}^g, q)$, where $q$ is agent's pose.
7:     Crop an egocentric map $\mathbf{M}^e$ from $\mathbf{M}^g$.
8:     *// Infer heuristic direction*
9:     Obtain K points $\mathcal{P} = \{p_i\}_{i=1}^K$ being $r$ meters away from agent's position at angles $\Theta = \{\theta_i | \theta_i = \frac{2\pi i}{K}\}_{i=1}^K$.
10:    Calculate geodesic distance from the agent to these points $\mathcal{D} = \{d_i | d_i = D(p_i)\}_{i=1}^K$ on $\mathbf{M}^g$.
11:    Select a index set of explorable point $\mathcal{I} = \{i \mid \mathbf{M}^g[p_i] = 0, d_i < \gamma\}$.
12:    Obtain heuristic direction $\theta^* = S(\{\theta_i, i \in \mathcal{I}\})$, where $S$ returns the closest direction to current camera direction.
13:    *// Sample action and update policies*
14:    Sample navigation action $a^n \sim \pi^n(\cdot|o)$ and camera action $a^c \sim \pi^c(\cdot|\mathbf{M}^e, \theta^*, a^n)$.
15:    Compute reward via Eq. (1).
16:    Update the navigation and camera policies via PPO.
17: **end while**

## 3.3 Reward Function for Camera Policy

We expect the camera policy to help agents explore environments more efficiently. We achieve this objective in two ways. On the one hand, we encourage the camera policy to follow the heuristic rules by awarding actions that move cameras toward the heuristic direction. This heuristic reward is defined as the reduction of $s$ in successive time steps, *i.e.,* $r_{\text{heuristic}} = s_{t-1} - s_t$. The $s \in [0°, 180°]$ represents the angle between the camera direction and the heuristic direction. On the other hand, because it is impractical to design a heuristic rule to cover all situations, we explicitly encourage the camera policy to explore more areas by an area reward $r_{\text{area}} = c_t - c_{t-1}$, which indicates the increase of explored area $c$ in successive time steps. The camera agent makes its own decision according to recommended direction and occupancy map information. In addition, to avoid constant camera moving and reduce energy consumption, we expect agents to execute camera motor for camera moving as less frequently as possible by introducing a turn-penalty reward $r_{\text{turn-penalty}} = \mathbb{1}[\text{turn-camera}]$, where $\mathbb{1}[\text{turn-camera}]$ equals to 1 when the agent actuates camera motor to move a camera and otherwise equals to 0. The detail about camera motor execution will be described in Section 3.4. To sum up, the reward for camera policy is as follows:

$$r = \alpha \cdot r_{\text{heuristic}} + \beta \cdot r_{\text{area}} - r_{\text{turn-penalty}}, \tag{1}$$

where $\alpha$ and $\beta$ are two scaling hyper-parameters. We train the camera policy to maximize such exploration reward using PPO [62]. In this way, the EXPO camera policy is encouraged to take into account heuristic rule-based decisions and to explore more areas by executing the camera motor as less frequently as possible.

## 3.4 Combination of Navigation and Camera Policies

**Navigation and camera actions execution.** We assume the camera is attached to a mobile robot platform using a revolute joint. The robot can move to a location using a wheel actuator and turn the camera using a camera motor simultaneously. The action space in this paper is based on real-world coordinates. The rotation angles for navigation and camera actions are set to the same. Thus, if both navigation and camera actions tend to move to the same direction, *e.g.,* turning the platform and camera to the left, we only need to actuate the robot wheel and do not need to actuate the camera motor.

**Incorporating camera policy into existing navigation methods.** Existing navigation methods can be mainly categorized into two types, namely modular SLAM-based [58, 72] and end-to-end learning-based [68, 70]. For the first type of navigation method, we use the existing navigation policy to infer a navigation action, which is then fed to the proposed camera policy to infer a camera

action. The paradigm is shown in Figure 2. In this case, it is worth mentioning that the camera policy coordinates with the navigation policy by conditioning the output of the navigation policy. The navigation policy decides navigation action by exploiting a progressively built map, which is built from historical camera policy output. For the end-to-end learning-based navigation method, the existing navigation policy typically contains the same recurrent network as our camera policy and uses it to predict navigation action. To better coordinate two kinds of policies, we use one recurrent network to predict both navigation action and camera action. Specifically, we feed this recurrent network both camera policy input (*i.e.,* map and heuristic direction) and navigation policy input (typically RGB-D images). Unlike the paradigm in Figure 2, the recurrent network in this case does not need the features of navigation action anymore because it predicts both types of actions at the same time. We use the summation of exploration reward in Equation 1 and navigation reward in existing methods (typically the distance reduced to goals) to evaluate the goodness of predicted navigation-camera actions. The paradigm is shown in Figure **??** in Supplementary Materials.

## 4 Experiments

### 4.1 Experimental Setup

**Task details.** We follow MultiON [68] to place multiple goal objects in the environment randomly. The objects are embodied with cylinders in different colors. In this way, We are free to decide the location and the number of goal objects to adjust the task difficulty. The agent succeeds in an episode if it calls FOUND action within a 1.5 meters radius of all objects in order. We call FOUND automatically when the agent is near the current goal object because we focus on evaluating the effectiveness of the agent finding objects and navigating to them. By default, we place three goal objects and denote it 3-ON task. We also show results of 2-ON and 1-ON in Supplementary Materials. We perform experiments on two photorealistic 3D indoor environments, *i.e.,* Matterport3D [7] and Gibson [71].

We follow the existing work [68] to evaluate the navigation success rate using **Success** and **Progress** metrics. Success indicates the percentage of successfully finished episodes, while Progress indicates the percentage of objects being successfully found. We also evaluate the navigation efficiency using **SPL** and **PPL**, which are short for Success weighted by Path Length and Progress weighted by Path Length, respectively. The weight is proportional to the navigation efficiency and is defined as $d/\bar{d}$, where $d$ is the length of the ground-truth path and $\bar{d}$ is the path length traveled by an agent.

**Implementation details.** For navigation and camera actions, a FORWARD action moves the agent forward by 0.25 meters and a TURN action turns by $30°$. The maximum episode time step is 500 because an agent with a global ground-truth map (*i.e.,* oracle agent) finishes more than 97% of episodes within 500 steps. Our camera policy tries to narrow the performance gap between such an oracle agent and the agent with a progressively built map. We set $K = 8$, $r = 2.4$, $\gamma = r \times 1.2 = 2.88$ in heuristic module empirically. We use a mapper that outputs the ground-truth of occupancy anticipation [58], because how to train a good mapper is orthogonal to our work. Reward scaling factors $\alpha$ and $\beta$ are set to 10 and 1 respectively such that three reward terms are in the same order of magnitude at initialization. We evaluate models for five runs using the same set of random seeds and report the mean results and standard deviation. More details are shown in Supplementary Materials.

### 4.2 Baselines

**Mapping + FBE [72]:** This SLAM-based navigation method breaks the problem into mapping and path planning. We use depth projection [21] for mapping and frontier-boundary-exploration (FBE) method [72] to select an exploration point. Once the built map covers goal objects, the agent navigates to them by taking deterministic actions [8] along the path planned by A* algorithm [34].

**OccAnt [58]:** This baseline is the same as the previous one except that we replace the depth projection with an occupancy anticipation neural network [58]. Such a network can infer the occupancy state beyond the visible regions. We assume the neural network is well-trained so that we use the ground-truth occupancy state within the field of view for experiments.

Table 1: Multi-object navigation results (%) for incorporating camera policy into different baselines on Matterport3D and Gibson datasets.

| Method | MatterPort3D | | | | Gibson | | | |
|---|---|---|---|---|---|---|---|---|
| | SPL | PPL | Success | Progress | SPL | PPL | Success | Progress |
| OccAnt | $53.0_{\pm0.0}$ | $57.7_{\pm0.1}$ | $72.0_{\pm0.1}$ | $80.2_{\pm0.1}$ | $76.1_{\pm0.1}$ | $77.8_{\pm0.1}$ | $89.0_{\pm0.1}$ | $91.8_{\pm0.1}$ |
| + Naive Camera Action | $48.7_{\pm0.5}$ | $53.1_{\pm0.4}$ | $69.1_{\pm0.6}$ | $76.8_{\pm0.2}$ | $69.9_{\pm0.6}$ | $71.9_{\pm0.4}$ | $84.8_{\pm1.1}$ | $88.3_{\pm0.8}$ |
| **+ Our Camera Policy** | $\mathbf{57.9}_{\pm0.5}$ | $\mathbf{62.1}_{\pm0.2}$ | $\mathbf{75.6}_{\pm0.5}$ | $\mathbf{82.6}_{\pm0.2}$ | $\mathbf{78.9}_{\pm0.5}$ | $\mathbf{80.7}_{\pm0.5}$ | $\mathbf{89.9}_{\pm0.5}$ | $\mathbf{92.4}_{\pm0.5}$ |
| Mapping + FBE | $40.1_{\pm0.0}$ | $45.4_{\pm0.1}$ | $62.3_{\pm0.0}$ | $72.5_{\pm0.2}$ | $62.1_{\pm0.3}$ | $64.9_{\pm0.2}$ | $80.9_{\pm0.4}$ | $86.2_{\pm0.4}$ |
| + Naive Camera Action | $35.2_{\pm1.1}$ | $41.2_{\pm1.0}$ | $55.3_{\pm1.4}$ | $66.9_{\pm1.3}$ | $55.8_{\pm0.5}$ | $58.5_{\pm0.4}$ | $77.5_{\pm0.6}$ | $83.1_{\pm0.3}$ |
| **+ Our Camera Policy** | $\mathbf{44.6}_{\pm0.3}$ | $\mathbf{49.8}_{\pm0.1}$ | $\mathbf{64.2}_{\pm0.9}$ | $\mathbf{74.1}_{\pm0.3}$ | $\mathbf{68.7}_{\pm0.1}$ | $\mathbf{71.2}_{\pm0.1}$ | $\mathbf{84.4}_{\pm0.1}$ | $\mathbf{88.9}_{\pm0.4}$ |
| MultiON | $33.0_{\pm0.5}$ | $43.8_{\pm0.6}$ | $44.1_{\pm0.6}$ | $60.5_{\pm0.5}$ | $56.5_{\pm0.3}$ | $62.2_{\pm0.2}$ | $68.4_{\pm0.1}$ | $77.3_{\pm0.0}$ |
| + Naive Camera Action | $32.4_{\pm0.1}$ | $45.1_{\pm0.2}$ | $41.9_{\pm0.1}$ | $60.2_{\pm0.8}$ | $54.1_{\pm0.3}$ | $61.6_{\pm0.0}$ | $64.5_{\pm0.5}$ | $75.2_{\pm0.2}$ |
| **+ Our Camera Policy** | $\mathbf{38.7}_{\pm0.7}$ | $\mathbf{49.5}_{\pm0.9}$ | $\mathbf{51.1}_{\pm0.0}$ | $\mathbf{67.3}_{\pm0.3}$ | $\mathbf{59.6}_{\pm0.3}$ | $\mathbf{66.8}_{\pm0.2}$ | $\mathbf{69.1}_{\pm0.3}$ | $\mathbf{79.0}_{\pm0.1}$ |
| DD-PPO | $16.7_{\pm0.2}$ | $29.2_{\pm0.2}$ | $22.2_{\pm0.1}$ | $40.9_{\pm0.3}$ | $30.1_{\pm0.4}$ | $41.2_{\pm0.4}$ | $39.4_{\pm0.7}$ | $54.8_{\pm0.5}$ |
| + Naive Camera Action | $16.7_{\pm0.1}$ | $30.4_{\pm0.1}$ | $20.8_{\pm0.3}$ | $39.2_{\pm0.4}$ | $31.3_{\pm0.3}$ | $43.4_{\pm0.2}$ | $38.8_{\pm0.6}$ | $54.5_{\pm0.5}$ |
| **+ Our Camera Policy** | $\mathbf{19.1}_{\pm0.4}$ | $\mathbf{34.0}_{\pm0.4}$ | $\mathbf{24.0}_{\pm0.2}$ | $\mathbf{43.9}_{\pm0.3}$ | $\mathbf{33.9}_{\pm0.3}$ | $\mathbf{45.3}_{\pm0.3}$ | $\mathbf{40.5}_{\pm0.3}$ | $\mathbf{55.3}_{\pm0.3}$ |

**DD-PPO [70]:** This end-to-end learning-based baseline performs navigation using RL algorithm. It consists of a recurrent policy network, which takes as input RGB-D images, goal object categories and previous actions for predicting navigation action.

**MultiON [68]:** This is the variant of DD-PPO, with a progressively built object map as an extra input. Each cell of the object map is a one-hot vector indicating the existence of the goal objects. We store an object on the map once it is within the field of view of agents. We encode the egocentric cropped object map and feed it to the policy network. This baseline is the same as *OracleEgoMap* variant presented by [68].

## 4.3 Multi-object Navigation Results

**Results on Matterport3D dataset.** In Table 1, the agent with our EXPO camera policy performs better on multi-object navigation task upon four baselines. Specifically, our agent increases Success and Progress metrics for a large margin, indicating incorporating our camera policy helps agents successfully navigate to more goal objects. We attribute the improvement to a better exploration ability of our agent. With such ability, the agent finds more goal objects in a limited time step and then navigates to them. Besides, the improvement on SPL and PPL indicates our agent navigates to goal objects along a shorter path. The agent does not need to walk inside all rooms. Instead, the actively moving camera allows the agent to perceive what is inside a room when it passes by. The above results show our agent navigates to goal objects more efficiently with a higher success rate. We encourage readers to see the visualization in Figure 4 and watch supplementary videos.

The camera action provides more movement freedom for the agent. We are interested in the question that whether the improvement comes from simply extending the action space. To this end, we remove our camera policy. The agent determines camera actions naively. Specifically, for SLAM-based baselines, the agent chooses a camera action randomly. For learning-based baselines, the agent learns to predict both camera and navigation actions using the original navigation input and rewards. In Table 1, using these naive camera actions brings little improvement or even negative influence. This is not surprising because it is nontrivial to coordinate the camera and navigation actions. Also, a larger action space may increase the learning difficulty. These results further suggest the importance of the proposed camera policy for determining a reasonable camera action.

**Transferability of camera policy to Gibson dataset.** We evaluate the transferability of the learned camera policy on Gibson dataset. In Table 1, a similar trend is observed on all baselines, *i.e.,* using naive camera actions does not help for navigation while our EXPO camera policy performs better than baselines. It is worth noting that the EXPO camera policy is trained on Matterport3D and has not been fine-tuned on Gibson dataset. There are significant differences between these two datasets in scene style and layout distribution [7, 71]. The consistent improvement demonstrates that our active-camera agent has learned general exploration and navigation skills for the multi-object navigation task. It also shows the possibility of transferring the agent to a real-world scene.

Table 2: Ablation study on input information and reward types of camera policy based on OccAnt baseline.

| Method | SPL | PPL | Success | Progress |
|---|---|---|---|---|
| w/o Map Input | $55.9_{\pm0.7}$ | $60.1_{\pm0.6}$ | $73.2_{\pm0.7}$ | $80.3_{\pm0.5}$ |
| w/o Heuristic Input | $56.5_{\pm0.6}$ | $60.4_{\pm0.4}$ | $74.0_{\pm0.3}$ | $80.7_{\pm0.2}$ |
| w/o NavAction Input | $56.9_{\pm0.4}$ | $60.7_{\pm0.3}$ | $74.7_{\pm0.5}$ | $81.2_{\pm0.3}$ |
| w/o Heuristic Reward | $53.8_{\pm0.5}$ | $58.1_{\pm0.3}$ | $72.7_{\pm0.7}$ | $80.3_{\pm0.4}$ |
| w/o Area Reward | $56.9_{\pm0.2}$ | $61.0_{\pm0.2}$ | $74.3_{\pm0.1}$ | $81.4_{\pm0.5}$ |
| w/o Turn Reward | $57.2_{\pm0.8}$ | $61.1_{\pm0.5}$ | $74.6_{\pm0.5}$ | $81.7_{\pm0.4}$ |
| **Ours** | $\mathbf{57.9}_{\pm0.5}$ | $\mathbf{62.1}_{\pm0.2}$ | $\mathbf{75.6}_{\pm0.5}$ | $\mathbf{82.6}_{\pm0.2}$ |

Table 3: Comparison between rule-based camera actions and learned camera policy based on two types of baselines.

| Method | SPL | PPL | Success | Progress |
|---|---|---|---|---|
| OccAnt | $52.9_{\pm0.0}$ | $57.7_{\pm0.1}$ | $72.0_{\pm0.1}$ | $80.2_{\pm0.1}$ |
| + Random Action | $48.7_{\pm0.5}$ | $53.1_{\pm0.4}$ | $69.1_{\pm0.6}$ | $76.8_{\pm0.2}$ |
| + Swing Action | $55.6_{\pm0.1}$ | $60.1_{\pm0.0}$ | $73.1_{\pm0.1}$ | $81.4_{\pm0.1}$ |
| + Heuristic Action | $56.3_{\pm0.1}$ | $60.3_{\pm0.1}$ | $74.0_{\pm0.1}$ | $80.9_{\pm0.1}$ |
| **+ Learned Policy** | $\mathbf{57.9}_{\pm0.5}$ | $\mathbf{62.1}_{\pm0.2}$ | $\mathbf{75.6}_{\pm0.5}$ | $\mathbf{82.6}_{\pm0.2}$ |
| MultiON | $33.0_{\pm0.5}$ | $43.8_{\pm0.6}$ | $44.1_{\pm0.6}$ | $60.5_{\pm0.5}$ |
| + Random Action | $3.1_{\pm0.6}$ | $10.2_{\pm0.5}$ | $4.8_{\pm0.9}$ | $19.8_{\pm0.1}$ |
| + Swing Action | $13.0_{\pm0.8}$ | $25.4_{\pm0.7}$ | $19.7_{\pm1.3}$ | $41.1_{\pm1.4}$ |
| + Heuristic Action | $19.9_{\pm0.7}$ | $32.2_{\pm0.4}$ | $27.6_{\pm0.8}$ | $46.3_{\pm0.4}$ |
| **+ Learned Policy** | $\mathbf{38.7}_{\pm0.7}$ | $\mathbf{49.5}_{\pm0.9}$ | $\mathbf{51.1}_{\pm0.0}$ | $\mathbf{67.3}_{\pm0.3}$ |

## 4.4 Further Analysis

**Rule-based camera actions *vs*. learned camera policy.** In contrast to learning a neural network, one may use handcrafted rules to decide camera actions. We compare the learned camera policy with three types of rule-based camera actions, *i.e.,* 1) selecting a random camera action; 2) forcing the agent to look forward and swing within $90°$ around the navigation direction; 3) following the heuristic direction inferred from the heuristic module.

The results upon a SLAM-based baseline, *i.e.,* OccAnt, are shown in Table 3. Exploiting random camera actions drops the performance because the agent often looks backward and captures redundant useless information. The other two types of rule-based camera actions improve the performance slightly. These camera actions help the agent build a map covering more areas. Consequently, the SLAM-based navigation policy can plan a better path for navigation using a path-planning algorithm. However, it is hard for us to design a robust rule covering all situations. For example, the swing camera actions may miss some explorable areas because the agent has passed by these areas before the camera swing to the direction pointing to them. Also, there exist false positive areas in the occupancy map (*e.g.,* free space behind a table is predicted as an obstacle). These areas may mislead the heuristic module to consider an unexplored area as an explored one. Compared with these camera actions, our learned camera policy brings a more significant improvement. We attribute the improvement to the exploration reward in Equation 1. With such a reward, the agent is encouraged to take into account not only the handcraft rules but also the noisy occupancy map to predict a better camera action.

As for learning-based baseline, *i.e.,* MultiON, using these three types of rule-based camera actions significantly drops the performance. In these experiments, it is worth noting that we have fed previous camera action to inform the policy in which direction the observation is taken from. We suspect the poor performance is because the camera movement decided by rules is uncontrollable by the navigation policy. As a result, the navigation policy can not get desired observations for predicting the next navigation action. In contrast, the learned camera policy, which is trained together with the navigation policy, allows the agent to determine how to move its camera by itself. The above results further demonstrate the importance of the learned EXPO camera policy.

**Ablation study on camera policy inputs and rewards.** We conduct this ablation study by removing one of the inputs and rewards of a camera policy upon OccAnt baseline on MatterPort3D dataset. In Table 2, removing any input or reward will drop the performance. We note that human knowledge (*i.e.,* heuristic input and reward) is important for the camera policy. Awarding the policy to follow this knowledge can be considered as a form of regularization and guidance for learning. The egocentric occupancy map input and area reward are also critical. They encourage the agent to explore more areas. With a better exploration ability, the agent can find goal objects and navigate to them more efficiently. The turn-penalty reward has little influence on navigation performance. However, it helps to reduce the frequency of actuating camera motor described in Section 3.4. Experimental results show that without this reward, agents actuate the camera motor for 12.41% of total time steps. Adding turn-penalty reward decreases the number to 5.70%.

**Does camera policy work with imperfect mapper?** Our EXPO camera policy obtains indoor layout information mainly from the progressively built map. In this subsection, we would like to evaluate whether the proposed camera policy works with a noisy map. To this end, we use a learned neural network [58] to predict the occupancy map from RGB-D images. The predicted map contains many false positive points (*e.g.,* predicting the free space as occupancy) and false negative points (*e.g.,*

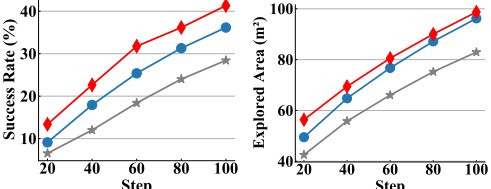

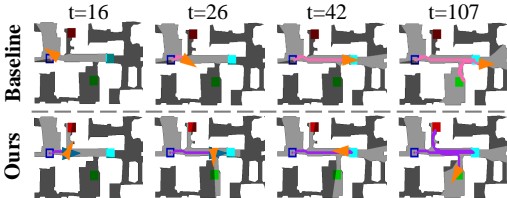

Figure 3: Exploration results. Left: success rate of finding three objects. Right: explored areas within limited time step. Standard deviation are negligibly small. -•-: w/o camera action. -★-: Random camera action. -◆-: Our camera policy.

Figure 4: Visualization of multi-object navigation. The start position: ☐ . Goal order: 1 ■, 2 ■, 3 ■. Camera direction: ▸. Navigation direction: ▸. When only ▸ is displayed, the camera and navigation directions are the same.

predicting a wall as free space). These noisy points may mislead the camera policy to make a wrong camera action. Experimental results on the OccAnt baseline show that incorporating camera policy brings consistent improvement. Specifically, SPL and PPL increase from 13.6 to 18.1 and from 20.5 to 25.4, respectively. Success and Progress increase from 18.39 to 24.0 and from 28.4 to 34.2, respectively. These results demonstrate that the proposed EXPO camera policy is robust to the map noise and can be deployed in a robot with no need for a ground-truth map.

**Does improvement come from extra information?** Compared with baselines, our active-camera agent leverages extra information (*i.e.,* occupancy map, heuristic direction and exploration reward) for determining camera actions. For a fair comparison, we add this extra information into MultiON to build an enhanced baseline. Results on MatterPort3D dataset demonstrate that the enhanced baseline performs slightly better than MultiON baseline but worse than our active-camera agent. Specifically, SPL, PPL, Success and Progress are 35.1, 45.9, 48.0, and 63.2, respectively. We suspect the improvement of the enhanced baseline comes from the fact that heuristic direction provides location information about unexplored areas. Also, the extra rewards encourage agents to explore these areas. However, due to the limited action space, the agent in enhanced baseline can not coordinate their camera and navigation actions well, which limits the performance.

**Exploration performance.** One of the critical abilities for the multi-object navigation task is exploring the environment efficiently to locate all goal objects. To evaluate the exploration ability, we place the agent in a novel environment with three goal objects located in different places. We follow FBE method [72] to explore the environment. Given a limited time step budget, we evaluate the success rate of finding all three goal objects and explored areas in Figures 3. Compared with the baselines that the agent is always looking forward or moving its camera randomly, the agent with our EXPO camera policy finds more goal objects and explores more areas. These results have the same trend of navigation performance in Table 1, suggesting that our EXPO camera policy helps to explore the environment more efficiently and consequently boosts the multi-object navigation performance.

**Visualization.** In Figure 4, both baseline (*i.e.,* OccAnt) and our agent are navigating to *Goal-1* at the beginning. During this process, our agent moves its camera actively at time step $t = 16$ and $t = 26$, finding *Goal-2* and *Goal-3* respectively. Knowing the location of goal objects, our agent plans the shortest path for navigation. In contrast, the baseline agent goes straight to *Goal-1*, with the camera looking forward constantly. Consequently, after it navigates to *Goal-1*, it cannot find other goal objects and has to waste time exploring the environment again. Failure case analysis can be found in Supplementary Materials.

## 5   Conclusion

In order to solve the uncoordinated camera-navigation actions problem of existing agents, we propose a navigation paradigm in which agents can dynamically move their cameras for perceiving environments more efficiently. Such exploration ability is important for multi-object navigation. To determine the camera actions, we learn a camera policy via reinforcement learning by awarding it to explore more areas. Also, we use heuristic rules to guide the learning process and reduce the learning difficulty. The proposed camera policy can be incorporated into most existing navigation methods. Experimental results show that our camera policy consistently improves the multi-object navigation performance of multiple existing methods on two benchmark datasets.

## Acknowledgments

This work was partially supported by National Key R&D Program of China (No. 2020AAA0106900), National Natural Science Foundation of China (No. 62072190 and No. 62006137), Program for Guangdong Introducing Innovative and Enterpreneurial Teams (No. 2017ZT07X183).

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
