# OpenReview forum: "Learning Active Camera  for Multi-Object Navigation"
_NeurIPS.cc/2022/Conference — NeurIPS 2022 Accept_

### Official Review · Reviewer_uqJh · 2022-07-10

**Rating:** 7
**Confidence:** 4
**Soundness:** 4 excellent
**Presentation:** 3 good
**Contribution:** 4 excellent

**Summary:**

In this paper, the authors propose to learn the active camera policy for multi-object robot navigation. Specifically, they show how to use camera rotations effectively to increase the information while in motion, and how to integrate the active camera policy with the navigation policy. Authors train active camera policy using Reinforcement Learning (RL) and carefully design a reward function that encourages robots to explore more areas. Through multiple baseline methods composed of SLAM-based, and end-to-end learning-based, the authors show that integrating active camera policy with the base navigation policy improves object navigation accuracy and efficiency on multiple metrics.

**Questions:**

1. What is the egocentric map size and how did you choose it? How does the policy performance change with increasing / decreasing this map size? I believe increasing cropped egocentric map would increase % of an unexplored area of the input map, and it would be good to see how it affects the policy.
2. For the experiment with an imperfect mapper, can the authors please also add what was the accuracy of such a trained mapper?


**Limitations:**

The authors have mentioned very generic limitations. Few assumptions of the paper should be part of the limitations as well.
1. Along with testing the effectiveness of such a method in the real-world environment, it is also useful to mention the limitation that different types of robots might also have to experiment with environment representations.
2. This active camera policy limits the use of a learning-based locomotion policy for high DoF robots such as Spot, and ANYMAL. Since the camera is not facing the robots' traversal path continuously.



**Strengths And Weaknesses:**

Strengths:
1. Paper is well written. The authors show clear motivation for the need for active cameras to aid exploration while navigating. They formulate the problem, propose a reinforcement-learning-based solution, and show the effectiveness of active cameras through extensive experiments.
2. Trained Active Camera Policy (EXPO) generalizes to other datasets than the one it’s been trained on (MatterPort3d → Gibson)
3. Authors show ablation study to show the usefulness of Active Camera Policy design components
4. Authors show improvements on 4 different baselines applied to 2 datasets. Baseline methods consist of SLAM-based, end-to-end learning-based methods, and hybrid methods.

Weakness:
1. Since the method is primarily trying to increase the scan horizon while moving, the authors should provide some details on the camera field of view. While I understand this is standard for the datasets like Mattterport3D, and Gibson, to be comprehensive it is necessary to include the field of view in the paper.
2. While an active camera is a great idea to scan an unexplored environment map while moving, the same can be achieved by fitting multiple cameras and ensuring we have a large field of view. Ideally, having multiple cameras covering a larger field of view will serve as an upper bound of what can be achieved.
3. I encourage authors to add more limitations of the method as described in the limitations review section

---

> ### Author Response · Authors · 2022-08-02
> **Respond to Reviewer uqJh**
>
> We thank you for your time and valuable comments. Below we answer the main concerns raised in the review and would be happy to provide further clarification if suitable.
>
>
> > **Q1) Since the method is primarily trying to increase the scan horizon while moving, the authors should provide some details on the camera's field of view.**
>
> Thanks for your suggestion. We follow the standard settings of the multi-object navigation task [a] and set the HFoV to 79°. We will add more crucial implementation details to the revised paper.
>
> > **Q2) While an active camera is a great idea to scan an unexplored environment map while moving, the same can be achieved by fitting multiple cameras and ensuring we have a large field of view.**
>
> Thank you for your recognition of our active camera.
>
> To compare our single active camera with a multi-fixed-camera agent, we equip an agent with four fixed pinhole cameras facing different directions (front, left, right, and back). Specifically, we concatenate the features of these four images and feed them to an end-to-end navigation baseline (i.e., OccAnt). We train this baseline for 30 million frames, which is the same as our active camera agent. The results are shown below:
>
> | Camera Type | SPL | PPL | Success | Progress |
> |:---|:---|:---|:---|:---|
> | Multi-Fixed-Camera | 14.4 | 29.1 | 17.9 | 38.7 |
> | Single-Active-Camera  (Ours) | **38.7** | **49.5** | **51.1** | **67.3** |
>
> From the results, our single-active-camera agent performs better than the baseline with four fixed cameras. We also observe that the multi-fixed-camera agent has not converged within 30 million frames. We speculate this is because the large amount of information from different directions increases the learning difficulty. More specific designs for using multi-fixed-camera are needed for further study.
>
> > **Q3)  I encourage authors to mention more limitations of the method. For example, different types of robots, especially the high DOF robots, are needed to be evaluated with active cameras.**
>
> Thanks for your valuable suggestion. Although we have evaluated the transferability of camera policy among different datasets (transferring from MatterPort3D to Gibson) in our paper, evaluating its robustness among different robot types and different embodied tasks is a valuable research direction. We will discuss these limitations and future work directions in the revised paper.
>
> > **Q4) What is the egocentric map size and how did you choose it? How does the policy performance change with increasing / decreasing this map size?**
>
> The egocentric map size is set to 125x125. We choose it by considering whether the map provides enough information for deciding actions. Specifically, we first set the map resolution as 0.08m, so that the map is fine-grained enough for path planning. Then, we have tried different map sizes, including 50x50, 75x75, 100x100, 125x125 and 150x150. The map with different map size cover different area of the environment. The results of our active camera agent based on OccAnt are as follows.
>
> | Map Size | SPL | PPL | Success | Progress |
> |:---|:---|:---|:---|:---|
> | 75x75 | 56.3 | 60.4 | 73.8 | 81.0 |
> | 100x100 | 56.5 | 61.1 | 73.4 | 80.8 |
> | 125x125 (Ours) | **57.9** | **62.1** | **75.6** | **82.6** |
> | 150x150 | 56.7 | 61.0 | 73.2 | 80.8 |
>
> From the results, an agent with a 125x125 map (which represents 10m x 10m around the agents) performs best. We observe that the map with a too small size cannot provide enough information for deciding navigation and camera actions. If the map is too large, it contains too much envrionment information far away from the agent, which may also deteriorate the performance.
>
> > **Q5) For the experiment with an imperfect mapper, can the authors please also add what was the accuracy of such a trained mapper?**
>
> We use the pre-trained Mapper from existing work [b], whose accuracy for local map prediction is 75.2%. More analysis and visualization results of this mapper can be seen in the original paper [b]. We will add these details in the revised paper.
>
> **Reference:**
>
> [a] MultiON: Benchmarking Semantic Map Memory using Multi-Object Navigation. In NeurIPS
>
> [b] Occupancy Anticipation for Efficient Exploration and Navigation. In ECCV

---

> > ### Comment · Reviewer_uqJh · 2022-08-08
> > **Thanks for further results**
> >
> > Thank you authors for taking time to providing further analysis and clarifications.
> >
> > Q 1) 3) 5) -- I think this information is good to fit in the main paper.
> >
> > Q 2) I am also surprised by the results of multi-fixed camera being this bad compared to active camera or even single camera policy. I see your analysis in response to reviewer pNpw. It would be nice to have a large scale experiment in the final version. One idea also could be to add position of the cameras (front, right, left, rear) and concatenate that with the image features for multi-camera policy.
> >
> > Q 4) Thanks for providing further experiment with different mapper input-size.

---

> > > ### Author Response · Authors · 2022-08-08
> > > **Thank you!**
> > >
> > > Thank you for your valuable reviews! We agree that these clarifications will make the paper better and will add them to the revised paper. Also thanks for your suggestion on multi-camera agents. We will consider it in a larger-scale experiment.

---

### Official Review · Reviewer_pNpw · 2022-07-10

**Rating:** 6
**Confidence:** 3
**Soundness:** 3 good
**Presentation:** 4 excellent
**Contribution:** 2 fair

**Summary:**

The paper tackles the challenge of SLAM-based multi-object navigation for indoor
robots. This task is typically approached with an algorithmic or RL method which
drives a robot equipped with an RGB-D sensor through an unknown finite
environment. The robot's goal is to find a series of predefined objects under a
limited step budget.

The paper makes the observation that prior approaches, such as Active Neural
SLAM, assume a fixed camera robot configuration with a limited forward-facing
FoV. The paper then proposes decoupling the robot motion from the camera yaw.
The robot can thus choose to, e.g., move in a direction while looking to the
side if it decides this provides more information. Two policies are learned,
one for navigation, one for looking around, with the navigation decision
feeding into the camera policy. The navigation policy is also responsible for
detecting the goal.

The paper focuses on the camera policy, which is designed to work both on top
of SLAM-based, as well as end-to-end navigation policies.

The method is trained on-policy using the photorealistic 3D environment
Matterport3D. The camera policy can generalize to the Gibson environment without
fine-tuning. In general, allowing the robot to move its camera independent of
its movement direction improves over baselines like not moving the camera,
moving it randomly, or moving it according to predefined patterns. This
improvement is consistent when evaluated on top of multiple navigation methods.
The paper also includes ablation studies for some of the key design decisions.


**Questions:**

- [Q1] What is the actuation noise in the environment? If the policy always expects
  turning itself or its camera in exact 30° increments, suddenly rolling out in an
  environment with noisy actuation may cause serious problems.


**Limitations:**

Please see "Weaknesses".

**Strengths And Weaknesses:**

## Strengths
- [S1] The paper demonstrates cross-domain generalization by training on
  Matterport3D and successfully transferring the policies to the Gibson Env
  without any fine-tuning.
- [S2] The proposed method is also shown to help improve both a SLAM-based, as
  well as a learning-based baseline.
- [S3] The paper defines an interesting new direction in active sensing.

## Weaknesses
- [W1] Clarity: It took me a while to realize that the method is actually fully
  compatible with both SLAM-based and end-to-end navigation systems. Would it be
  possible to add a diagram, maybe in the supmat if the main paper can't fit it,
  showing the specific information flow used when the end-to-end paradigm is
  used in navigation?
  - Either way, it would be nice to clarify this earlier in the paper, as it
    makes it easier to understand how the camera policy fits into the existing
    navigation pipeline.
- [W2] Multi-Camera Baseline: While I do understand the point brought up in
  Appendix A, I am not fully convinced by the argument.
  - A multi-pinhole-camera robot could make for an interesting comparison, in
    spite of the increased costs. A complex panoramic camera (with the
    associated geometric and hardware complexity) is not strictly required.
  - An additional advantage of multiple fixed camera is eliminating the need for
    the moving parts of the revolute joint and its associated servo.
  - Second, I wouldn't call the computational overhead of, say, four cameras
    "huge", since naive baselines like concatenating the images before feeding
    them to an end-to-end navigation policy would probably work pretty well
    already.
  - Multi-camera systems, albeit not depth-based, are already ubiquitous in mobile
    phones, drones, and (even non-autonomous) vehicles.
  - That being said, I still think it makes sense for this problem setting to
    exist, as coordinating between motion and active sensing is generally an
    important problem in robotics, which hasn't received as much attention as it
    should have.


## Suggestions
- L194: "module SLAM-based" -> "modular, SLAM-based"?

---

> ### Author Response · Authors · 2022-08-02
> **Respond to Reviewer pNpw**
>
> We thank you for your time and valuable comments. Below we answer the main concerns raised in the review and would be happy to provide further clarification if suitable.
>
> > **Q1) Would it be possible to add a diagram, showing the specific information flow used when the end-to-end paradigm is used in navigation?**
>
> Thanks for your valuable suggestion. We will make it more clear by adding an information flow diagram for the end-to-end paradigm in the supplementary.
>
> > **Q2) Multi-Camera Baseline: A multi-pinhole-camera robot could make for an interesting comparison. That being said, I still think it makes sense for the active camera setting to exist, as coordinating between motion and active sensing is generally an important problem in robotics, which hasn't received as much attention as it should have.**
>
> Thank you for your recognition of our active camera.
>
> To compare our single active camera with the multi-fixed-camera agent, we equip an agent with four fixed pinhole cameras facing different directions (front, left, right, and back). Specifically, we concatenate the features of these four images and feed them to an end-to-end navigation baseline (i.e., OccAnt). We train this baseline for 30 million frames, which is the same as our active camera agent. The results are shown below:
>
> | Camera Type | SPL | PPL | Success | Progress |
> |:---|:---:|:---:|:---:|:---:|
> | Multi-Fixed-Camera | 14.4 | 29.1 | 17.9 | 38.7 |
> | Single-Active-Camera  (Ours) | **38.7** | **49.5** | **51.1** | **67.3** |
>
> From the results, our single-active-camera agent performs better than the baseline with four fixed cameras. We also observe that the multi-fixed-camera agent has not converged within 30 million frames. We speculate this is because the large amount of information from different directions increases the learning difficulty. More specific designs for using multi-fixed-camera are needed for further study.
>
> > **Q3) What is the actuation noise in the environment? If the policy always expects turning itself or its camera in exact 30° increments, suddenly rolling out in an environment with noisy actuation may cause serious problems.**
>
> Thanks for your valuable comment. In this paper, we follow the existing works [b, c] to simplify the problem by training and testing without considering noise. However, we agree that actuation and sensor noises are unavoidable in the real world.
>
> To respond to the Reviewer's concerns, we also evaluate our agents, which are trained in a noise-free environment, on a noisy environment. Specifically, we leverage the noise simulator in ANS [d] to simulate actuation noise and sensor noise for both body and camera movements. The results based on the MultiON baseline are as below (the absolution improvement are shown in the bracketed):
>
> * Evaluation without Noise:
>
> | Method | SPL | PPL | Success | Progress |
> |:---|:---|:---|:---|:---|
> | MultiON | 33.0 | 43.8 | 44.1 | 60.5 |
> | MultiON + Active Camera (Ours) | **38.7 (+5.7)** | **49.5 (+5.7)** | **51.1 (+7.0)** | **67.3 (+6.8)** |
>
> * Evaluation with Noise:
>
> | Method | SPL | PPL | Success | Progress |
> |:---|:---|:---|:---|:---|
> | MultiON | 3.1 | 9.4 | 6.7 | 22.5 |
> | MultiON + Active Camera (Ours) | **17.1 (+14.0)** | **25.2 (+15.8)** | **35.4 (+28.7)** | **56.9 (+34.4)** |
>
> From the results, the noise drops the performance of both our active camera agent and baseline agent. However, even under the noise evaluation setting, our active camera agent performs better than the baseline. We also observe that the agent with an active camera performs more robust under the noise condition.
>
> In this work, we focus on proposing a camera policy that helps to actively move the camera. To mitigate the effect of noise, there are many existing techniques, including training a neural pose estimator [d] and designing specific rewards for the RL process [a]. These techniques are proven to be effective for navigation tasks. Incorporating these noise mitigation techniques with our camera policy is an interesting research direction. We will leave it for future work.
>
>
> **Reference:**
>
> [a] Occupancy Anticipation for Efficient Exploration and Navigation. In ECCV
>
> [b] MultiON: Benchmarking Semantic Map Memory using Multi-Object Navigation. In NeurIPS
>
> [c] Cross-modal Map Learning for Vision and Language Navigation. In CVPR
>
> [d] Learning to Explore using Active Neural SLAM. In ICLR

---

> > ### Comment · Reviewer_pNpw · 2022-08-06
> > **Thank you for the follow-up analysis!**
> >
> > * **Multi-camera**: Thank you for the experiments. I am a bit puzzled regarding the poor performance of the multi-camera setup. It seems to perform very poorly, much worse than a front-facing camera, which does not make sense. Could it be a bug in the code, or do you think it's just due to the policy not having converged yet? It would be interesting to see a larger-scale experiment, especially based on a SLAM method, and see if the trend continues. For e2e policies some positional encoding may be necessary in the inputs in order to allow the robot to properly map view directions to actions.
> > * **Clarity**: Thank you for addressing the clarity question. The diagram should help improve the paper's clarity.
> > * **Actuation noise**: Thanks for including this experiment as well. It's worth adding it to the full paper. It seems that also applying noise at train time has potential to improve robustness!

---

> > > ### Author Response · Authors · 2022-08-07
> > > **Response to Reviewer pNpw**
> > >
> > > Thank you for the followup and comments! We're pleased that our response addresses some of your concerns, and hope to address the remainder now.
> > >
> > > > **Concerns on multi-camera results.**
> > >
> > > We totally agree with the Reviewer that positional encoding is important for the e2e baseline. Actually, in our experiments, we have concatenated each RGBD feature with its corresponding heading embedding. We also have carefully checked our code and believe the code runs as expected.
> > >
> > > **We speculate the worse performance is mainly caused by non-convergence.** As shown in the tables below, when we continue to train the models, the performance of the multi-camera agent continuously increases （Table A） while the performance of front-facing single-camera agent does not change a lot (Table B). We speculate the multi-camera agent may need more training samples for learning how to map the relatively large amount of information from different directions to correct actions. Due to the time limitation of rebuttal, we will conduct a larger scale experiment after rebuttal and update the results in the revised paper.
> > >
> > > * Table A: Results of e2e-based multi-fixed-camera agents (the absolution performance difference from 30M are shown In parentheses).
> > >
> > > |      Camera Type     | # Training Samples | SPL | PPL | Success | Progress |
> > > |:--------------------:|:--------------------:|:---:|:---:|:-------:|:--------:|
> > > |  Multi-Fixed-Camera  |         30M        |     14.4    |     29.1    |     17.9    |     38.7    |
> > > |  Multi-Fixed-Camera  |         33M        | 15.8 (+1.4) | 31.5 (+2.4) | 19.0 (+1.1) | 40.0 (+1.3) |
> > > |  Multi-Fixed-Camera  |         36M        | 17.3 (+2.9) | 32.8 (+3.7) | 20.2 (+2.3) | 41.1 (+2.4) |
> > >
> > >
> > > * Table B: Results of e2e-based single-fixed-camera agents (the absolution performance difference from 30M are shown In parentheses
> > >
> > > |      Camera Type     | # Training Samples | SPL | PPL | Success | Progress |
> > > |:--------------------:|:--------------------:|:---:|:---:|:-------:|:--------:|
> > > | Single-Fixed-Camera |         30M        |     33.0    |     43.8    |     44.1    |     60.5    |
> > > | Single-Fixed-Camera |         33M        | 33.1 (+0.1) | 44.6 (+0.8) | 43.2 (-0.9) | 60.4 (-0.1) |
> > > | Single-Fixed-Camera |         36M        | 33.7 (+0.7) | 44.2 (+0.4) | 43.8 (-0.3) | 60.8 (+0.3) |
> > >
> > >
> > > **We also have conducted multi-camera experiments on the SLAM-based method.** The information captured by multiple cameras helps agents build a map efficiently. This map provides more information for path-planning. The results are shown below (Table C). The multi-camera agent outperforms our active camera agent. Because the SLAM-based multi-camera agent does not need to learn a neural network to map RBGD observation to navigation actions, it will not suffer from the slow convergence problem as in e2e-based agents. It is worth noting that although using a single camera, our active camera policy helps the agent achieve comparative performance compared with an agent with four fixed cameras.
> > >
> > > * Table C: Comparisons of SLAM-based agents with different camera types.
> > >
> > > | Camera Type | SPL | PPL | Success | Progress |
> > > |:---:|:---:|:---:|:---:|:---:|
> > > |   Multi-Fixed-Camera | 68.3 | 72.2 | 79.4 | 85.3 |
> > > | Single-Active-Camera | 57.9 | 62.1 | 75.6 | 82.6 |
> > >
> > > > **Actuation Noise.**
> > >
> > > We totally agree with the Reviewer that applying noise at train time has the potential to improve robustness. We will add the actuation noise experimental results in the revised paper and leave the noise mitigation study for future work.
> > >
> > > Please don’t hesitate to let us know if there are any additional clarifications or experiments that we can offer. We appreciate your suggestions. Thanks!

---

> > > > ### Author Response · Authors · 2022-08-10
> > > > **Further study on actuation noise**
> > > >
> > > > As suggested by the Reviewer, applying noise at train time has the potential to improve robustness in a noisy evaluation environment. We totally agree with this idea. To evaluate its effectiveness, we conduct an experiment where we train and evaluate the agents in an environment with actuation noise and pose sensor noise. The evaluation results in a noisy environment for a MultiON-based active-camera agent are shown in the table below.
> > > >
> > > >
> > > > | Train w/ Noise | SPL  | PPL  | Success | Progress |
> > > > |----------------|------|------|---------|----------|
> > > > | No              | 17.1 | 25.2 | 35.4    | 56.9     |
> > > > | Yes             | 31.0 | 39.3 | 48.8    | 66.0     |
> > > >
> > > > From the results, simply applying noise at train time significantly increases the performance. This provides a simple but effective method for us to mitigate the effect of actuation and sensor noise. We believe applying other more complex noise estimation techniques could further improve the robustness.

---

### Official Review · Reviewer_iN1f · 2022-07-11

**Rating:** 6
**Confidence:** 4
**Soundness:** 2 fair
**Presentation:** 3 good
**Contribution:** 2 fair

**Summary:**

This paper enables efficient multi-goal navigation through active perception.
The camera of the agent, which determines its limited field of view, is actively controlled to turn towards scene regions that are still unexplored.
Camera rotation is decoupled from agent movement, i.e. the agent can still move in directions other than the facing direction.
The camera control problem is framed as an MDP with rewards proportional to spatial exploration of the environment (measured in uncovered area, based on a maintained 2D occupancy map).
Camera actions are produced by a parametric policy optimized through PPO for the aforementioned MDP (with an explicit critic that approximates state values).
The policy is informed through a very strong heuristic for how the camera should turn, in the form of an extra objective term, but also seems to improve upon it (due to the refinement through the exploration objective from the described MDP).
It is demonstrated that actively controlling the camera leads to improved efficiency and success rate for the multi-goal navigation task on the MatterPort3D and Gibson data sets.
Both SLAM-based and end-to-end RL navigation baselines are augmented with the active camera policy in the experiments.
Generalization from MatterPort3D (used for training) to Gibson is demonstrated.

**Technical details**
The proposed scheme is used to augment existing navigation methods that are already tailored to the multi-goal navigation task.
The camera policy is recurrent and is conditioned on features from a top-down 2D map.
It also takes in the hand-crafted heuristic camera control and the action produced by the movement policy as input (the latter is necesary only when a SLAM model is used, as camera and movement controllers are decoupled).
When an end-to-end RL model is considered, the camera and movement policy are modelled by the same recurrent neural network, jointly optimizing an MDP with rewards for both navigation and exploration.

**Questions:**

**Questions**
- The proposed method maintains a 2D occupancy map of the environment, which is necessary to compute the exploration heuristic. Where do the camera poses come from in the end-to-end learned RL experiments (for the SLAM-based navigation task I assume the SLAM estimates are used)? These poses are needed to update the occupancy map with the RGB-D observations.
- Have you considered quantifying the exact way in which the trained camera policy deviates from the heuristic?


**Limitations:**

I don't see any major negative societal impact. There is a short section on limitations right before the conclusion that mentions the lack of real-world experiments. It could be extended based on the remarks above.

**Strengths And Weaknesses:**

**Strengths**
- The approach appears straightforward & functional. The design choices are pragmatic.
- The motivation is easy to understand and follow.
- The fact that camera and motion control needs to be considered jointly has been accounted for, albeit only for the end-to-end learned systems.
- Ablations of the method are appreciated, they indicate that the different modelling choices were necessary.
- It is nice to see the policy generalizes from one data set to the other.

**Weaknesses**
- The authors discuss the camera policy in the context of an MDP, but are the agent states actually observed? My impression was that the multi-goal navigation task is a partially-observed problem, where the observations are only RGB-D images. In that case the ideal decision problem would technically be a POMDP, and this should be discussed in the manuscript.
- Optimizing the camera policy in isolation of the overarching movement control problem can be problematic, as the authors have also noted in the experimental section. This has been accounted for in the end-to-end learned experiments, where a joint control task is formulated. But for the SLAM-based controllers, I am not convinced conditioning on the movement controls is enough. Particularly if you consider the ideal POMDP formulation, the movement controller should be aware of the images that will be perceived in the future (due to camera control), as this will affect the quality of state estimation (which both policies should account for). What is the reason to keep the controllers separated in the SLAM case?
- In a POMDP the uncertainty of the state estimates normally plays a role, and I believe the agent states used to update the occupancy map are deterministic.
- The method is evaluated in only two environments in simulation, real-world experiments have remained out of scope.

Overall, I find that the proposed solution has value, the proposed idea is straightforward and seems to work as intended.
The solution is very specific in scope, tailored to the problem of actively controlling a camera for exploration, but on the other hand spatial exploration is an important topic and spatial agents are ubiquitous.
The other reservations I have are related to the missing discussion of how the method fits into the POMDP paradigm; I do not necessarily expect the method to change, but being explicit about its limitations would be appreciated.

---

> ### Author Response · Authors · 2022-08-02
> **Respond to Reviewer iN1f**
>
> We thank you for your time and valuable comments. Below we answer the main concerns raised in the review and would be happy to provide further clarification if suitable.
>
> > **Q1) Are the agent states actually observed? The ideal decision problem would technically be a POMDP.**
>
> Thank you very much for pointing it out. We totally agree with you that the agent's observations cannot fully specify the state of the environment in navigation tasks. In this sense, formulating the multi-goal navigation as a POMDP is more accurate. We will revise and discuss the formulation carefully in the revised paper.
>
>
> > **Q2) For the SLAM-based controllers, I am not convinced conditioning on the movement controls is enough.  What is the reason to keep the controllers separated in the SLAM case?**
>
> We agree that optimizing two policies as a joint control task enables coordination between two policies. We also want to point out that, in the SLAM case, the two policies are not totally separated for the following reasons.
>
> 1) The **camera policy** decides camera actions conditioning on the output of the navigation policy so that it is not separated from the navigation policy.
> 2) The **navigation policy** decides navigation action conditioning on a progressively built map, which is built from historical camera policy output. In this sense, the navigation policy is not separated from the camera policy. It is worth noticing that the SLAM-based navigation policy works well (i.e., planning the shortest path) without the need of knowing current camera actions. This is because the path planning destination always appears within the explored area of the map built in the last time step. In this sense, the images perceived in the future do not change the planned navigation actions.
>
> To further alleviate the concerns of the Reviewer, we have conducted experiments using a joint action space in the SLAM case. In the table below, this variant performs slightly worse than ours. We speculate this is because the larger joint action space increases the learning difficulty. We will add these discussions in the revised paper.
>
>
> |  | SPL | PPL | Success | Progress |
> |:---:|:---:|:---:|:---:|:---:|
> | SLAM-based Joint Actions | 52.0 | 56.5 | 71.4 | 79.0 |
> | SLAM-based (Ours) | **57.9** | **62.1** | **75.6** | **82.6** |
>
>
>
> > **Q3) In a POMDP the uncertainty of the state estimates normally plays a role, and I believe the agent states used to update the occupancy map are deterministic.**
>
> We agree that the uncertainty of the state estimates is important in a POMDP problem. In our paper, although we have not estimated the state uncertainty explicitly, we exploit a recurrent neural network to summarize the historical partial observations to better estimate the underlying system state. The recurrent neural network is proven to be effective to solve POMDP problems [a, b]. In addition, existing navigation works also demonstrate their effectiveness in solving partially observed navigation problems [c, d].
>
> Even so, we still believe the performance of our active camera agent can be further improved if we design a specific mechanism for estimating state uncertainty. We leave it for future work.
>
> > **Q4) The method is evaluated in only two environments in simulation, a real-world experiments have remained out of scope.**
>
> Following standard settings in visual navigation[c, d], we have evaluated our method in two benchmark simulated environments. Besides, we have evaluated the transferability ability of our method, i.e., training in one environment and testing in another environment. We agree that adapting the active camera policy to a real-world robot is an interesting research direction and we leave it for future work.
>
> > **Q5) Where do the camera poses come from in the end-to-end learned RL experiments (for the SLAM-based navigation task I assume the SLAM estimates are used)?**
>
> We follow standard settings in MultiON [c] that equip agents with a perfect camera pose sensor to simplify the problem. There exist many works for estimating pose [d, e] and achieving good performance on navigation. Our main focus is to propose a camera policy that helps to move the camera actively during the navigation process. Exploring pose estimation methods is out of scope in this paper.

---

> > ### Author Response · Authors · 2022-08-02
> > **Respond to Reviewer iN1f**
> >
> > > **Q6) Have you considered quantifying the exact way in which the trained camera policy deviates from the heuristic?**
> >
> > Thank you for your valuable suggestion. Quantitatively, we summarize the frequency that the trained camera policy deviates from the heuristic: 6.8% for SLAM-based baseline (i.e., OccAnt), 52.9% for end-to-end baseline (i.e., MultiON). These deviations bring 1.6% and 23.5% improvement in terms of success rate (as shown in Tab. 3 in the paper), respectively.
> >
> > From these results, we notice that the deviation occurs more frequently for end-to-end baseline, and brings more significant improvement. We speculate this is because the observation obtained by heuristic camera actions cannot provide desired information for end-to-end learned navigation policy for deciding navigation actions. In this sense, the deviation occurs frequently. By contrast, the SLAM-based navigation policy decides actions by performing a path-planning algorithm on a progressively built map. This eases the demand for controlling camera action to get specific observations (as discussed in the second point in Q2).
> >
> > > **Q7) The solution is very specific in scope, tailored to the problem of actively controlling a camera for exploration, but on the other hand spatial exploration is an important topic and spatial agents are ubiquitous.**
> >
> > Thanks for your comment. The main purpose of our paper is to propose an active camera policy that can cooperate with most existing navigation policies (i.e., SLAM-based and end-to-end policies). Experimental results show consistent improvements in these two types of mainstream navigation policies. Exploring a specific navigation policy that can better cooperate with our proposed camera policy for spatial exploration is a good research direction but out of scope in this paper.
> >
> > **Reference:**
> >
> > [a] Deep Recurrent Q-Learning for Partially Observable MDPs. In AAAI.
> >
> > [b] Memory-based Control with Recurrent Neural Networks. In NeurIPS
> >
> > [c] MultiON: Benchmarking Semantic Map Memory using Multi-Object Navigation. In NeurIPS
> >
> > [d] Occupancy Anticipation for Efficient Exploration and Navigation. In ECCV.
> >
> > [e] Learning to Explore using Active Neural SLAM. In ICLR

---

> > > ### Comment · Reviewer_iN1f · 2022-08-07
> > > **Thank you for your response**
> > >
> > > Thank you for taking the time to respond to my comments.
> > >
> > > I appreciate that you explicitly quantified the deviation of the learned policy from the heuristic, I believe this defines the merit of the proposed approach more clearly. In particular, it seems like the heuristic on its own is very strong in SLAM contexts already, and navigation based on end-to-end learning is primarily where the proposed learned active policy has an edge.
> > >
> > > In terms of state estimation and a POMDP treatment, I understand that the paper's main contribution is orthogonal to devising state estimators. Currently the method assumes perfect poses are available for end-to-end learning for MultiON, or it assumes the state estimates from SLAM are perfect. I agree that this is pragmatic from a research standpoint, allowing you to focus on the active data acquisition alone. It might be a good idea to highlight this aspect in the final version.
> > >
> > > The motivation for my original comments was that imperfect state estimation will inevitably arise on real hardware (which you mentioned is out of scope for now), and then the effect of future observations might have to be accounted for by the policy. E.g. the active camera policy & the navigation policy might need to learn to avoid regions that can throw off the state estimator. State estimation uncertainty would also matter in that case.

---

> > > > ### Author Response · Authors · 2022-08-08
> > > > **Thank you!**
> > > >
> > > > Thank you for your valuable reviews. We agree that these clarifications will make the paper better. We will add them to the revised paper.

---

### Author Response · Authors · 2022-08-02
**General Response to All Reviewers and ACs**

We sincerely appreciate all reviewers’ time and efforts in reviewing our paper and for the constructive feedback. In addition to the response to specific reviewers, here we would like to thank reviewers for their acknowledgment of our work and highlight the new results added during the rebuttal:

**We are glad that the reviewers appreciate and recognize our contributions:**

* The proposed solution has value [iN1f]
* Coordinating between motion and active sensing is an important problem in robotics [pNpw]
* An active camera is a great idea [uqJh]
* The paper is well written and easy to follow [iN1f, uqJh]
* Trained active camera policy generalizes to the other datasets well on both SLAM-based and learning-based settings [iN1f, pNpw, uqJh]

**In this rebuttal, we have added more supporting results following the reviewers’ suggestions.**

* Evaluate the SLAM-based method under joint action space [iN1f]
* Summarize the frequency that the trained camera policy deviates from the heuristic camera policy [iN1f]
* Compare our single active camera with the multi-fixed-camera agent [pNpw, uqJh]
* Evaluate our agents, which are trained in a noise-free environment, on a noisy environment [pNpw]
* Conduct ablation study on different map sizes [uqJh]


**We also address the reviewers’ questions by adding the following revision to the revised paper.**

* We reformulate the problem as POMDP in Section 3.1 [iN1f]
* Detailed analysis of how two policies coordinate in the SLAM case is added in Section 3.4 [iN1f]
* An information flow diagram for the end-to-end paradigm is added in the revised supplementary material [pNpw]
* Some crucial implementation details, including the camera FoV, are added in the revised supplementary material [uqJh]
* More limitations and future works are discussed in Section 5 [uqJh]

---

### Meta-Review · Area_Chair_J11v · 2022-08-20

**Recommendation:** Accept
**Confidence:** Certain

**Metareview:**

This paper proposes to decouple the camera policy from the navigation policy in goal-driven navigation agents trained using RL, and builds upon the local and global mapping and planning approach by adding an additional recurrent network that takes as inputs global reconstructed maps, heuristic directions, and navigation actions, to predict camera left/none/right turn actions. Rewards for the camera policy come from a camera turning heuristic and from map exploration heuristic. The agent is evaluated on multi-goal navigation tasks and tested and transferred to Matterport 3D and Gibson. The authors conduct a large set of ablations and comparison b/w different mapping and planning, SLAM-based or deep RL methods.

Reviewers praised the clarity and motivation of the method (iN1f, pNpw, uqJh), the ablations (iN1f, uqJh), evaluation and improvement on a SLAM baseline (pNpw, uqJh) and the generalisation performance (iN1f, pNpw, uqJh).
Reviewers noted that camera policy was optimized irrespective of future navigation strategies, which could be an issue (iN1f), and a lack of discussion about POMDP formulations of the navigation policy and assumptions about perfect state estimation (iN1f), limitations to 2D motion (uqJh), and recommended that the authors emphasize that the method can work with pure SLAM mapping (pNpw). The authors added experiments demonstrating that a single active camera outperformed non-active multi-camera systems (pNpw, uqJh), experiments on different sizes of the egocentric map in the planner (uqJh) and experiments with actuator noise (pNpw).

Reviewers agree on high scores (6, 6, 7) and therefore I would recommend this paper for acceptance.

Thank you,
Sincerely,
Area Chair


**Award:**

No

---

### Decision · Program_Chairs · 2022-09-14

Accept